# A new formulation of compartmental epidemic modelling for arbitrary distributions of incubation and removal times

**Pilar Hernández****[1]\*, Carlos Pena[2], Alberto Ramos[3], Juan José Gómez-Cadenas[4]**

**1** Departamento de Física Teórica e Instituto de Física Corpuscular CSIC-UVEG, Universidad de Valencia, Valencia, Spain, **2** Departamento de Física Teórica and Instituto de Física Teórica UAM-CSIC, Universidad Autónoma de Madrid, Madrid, Spain, **3** School of Mathematics & Hamilton Mathematics Institute, Trinity College Dublin, Dublin, Ireland, **4** Donostia International Physics Center (DIPC) and Ikerbasque, Donostia-San Sebastian, Spain

\* m.pilar.hernandez@uv.es

**Data Availability Statement:** The data in our figures can all be found in the OSF public repository: https://osf.io/cdakv/files/ including the data from the microdynamical simulations (ABM

## Abstract

The paradigm for compartment models in epidemiology assumes exponentially distributed incubation and removal times, which is not realistic in actual populations. Commonly used variations with multiple exponentially distributed variables are more flexible, yet do not allow for arbitrary distributions. We present a new formulation, focussing on the SEIR concept that allows to include general distributions of incubation and removal times. We compare the solution to two types of agent-based model simulations, a spatially homogeneous one where infection occurs by proximity, and a model on a scale-free network with varying clustering properties, where the infection between any two agents occurs via their link if it exists. We find good agreement in both cases. Furthermore a family of asymptotic solutions of the equations is found in terms of a logistic curve, which after a non-universal time shift, fits extremely well all the microdynamical simulations. The formulation allows for a simple numerical approach; software in Julia and Python is provided.

## Introduction

A burgeoning number of papers attempting to model the dynamics of the COVID-19 pandemic have been published over the last few months [1]. Among these, a large fraction ([2–4] are just a few examples) are based in more or less complex variants of the classical SEIR (susceptible-exposed/preinfectious-infectious-removed) model [5], which assume exponential distributions of incubation and removal times. Real epidemic data do not support however an exponential distribution for these parameters which are usually described by gamma, lognormal or Weibull distributions [6, 7].

It is well-known that these more general distributions are difficult to be captured by classical compartmental models such as SEIR, which normally treat the incubation and removal times as exponentially distributed, or as the sum of several exponentially-distributed independent times. On the other hand, there is a vast literature on the study of non-Markovian

and networks) as well as a Mathematica notebook that constructs the figures. The software that we have implemented to solve the uSEIR equations in the most general case is available in Julia and Python and can be downloaded from https://gitlab.ift.uam-csic.es/alberto/useir https://github.com/jjgomezcadenas/useirn.

**Funding:** The author(s) received no specific funding for this work.

**Competing interests:** The authors have declared that no competing interests exist.

stochastic processes to model epidemics [8], but no simple and concrete formulation of compartmental models for epidemics that implements general distributions. The goal of this paper is to provide such a formulation, which is also practical from a numerical point of view.

The principles on which a completely general formulation of compartmental models can be built are actually already present in the seminal work by Kermack and McKendrick [9], where they considered the SIR model. Here, we focus on the more general SEIR model. Our starting point is a formulation of the SEIR concept that correctly represents the evolution of an epidemic under the assumption of full homogeneity and fixed values of the microscopic parameters, namely: i) number of contacts per unit time, ii) the probability of infection per contact, iii) the incubation time and iv) the recovery/removal time. By construction the inter-compartment probabilistic transition rates transparently conserve the total probability as time evolves: we thus refer to our construction as uSEIR, where "u" stands for unitary.

We demonstrate that the results of uSEIR describe very well the result of a microscopic Agent Based Model (ABM) simulation with no stochasticity in the model parameters. Arbitrary distribution of the incubation and removal times can be easily incorporated into our equations, recovering the classical SEIR equations in the particular case of exponentially-distributed incubation and removal times. We also show that the resulting equations can be efficiently solved numerically, and provide appropriate codes and examples (with implementations in the Julia and Python/Cython languages).

The non-homogeneity in the infection rate per unit time is more subtle, because, in the extreme case, it should invalidate the treatment in terms of global S-E-I-R populations. We study two sources of this non-uniformity: inhomogeneity in the probability of infection per contact and inhomogeneity in the number of contacts per individual. The first is modelled with an ABM model with a negative binomial distribution of the probability of infection per contact, the second is modelled with a simulation on a scale-free network. The uSEIR equations represent instead the evolution in which the infection rate per unit time is the average one, independently of the underlying distribution. We observe that the simulations show a significant variance which however amounts mostly to a time-translation. When the different curves of infected individuals are shifted to tune their maxima, all the curves fall on a universal curve correctly reproduced by the uSEIR equations. We analyse the origin of this universality, i.e., independence on initial conditions. It derives from an asymptotic solution of the uSEIR equations, which is found to be a logistic curve whose shape is fixed by the average microscopic parameters. In the case of networks we briefly comment on the effect of clustering on the dynamics of the epidemic.

## Epidemic dynamics: From local to global

The spread of an infectious agent in a large population is a complex stochastic process, which under certain assumptions can be described in terms of a relatively small number of global variables, which follow deterministic differential equations. In order to understand the underlying dynamics, it is useful to think of an epidemic outbreak in terms of simple agent-based models (ABMs) [10], where the microdynamics can be studied by computer simulations. In these models agents can make their own decisions based on the rules given to them, and the evolution can capture unexpected aggregate phenomena that result from combined individual behaviours. ABMs can incorporate easily stochastic parameters as well as heterogeneities in the population, and they are therefore a useful tool to study the performance of the description in terms of global variables.

In the context of an epidemic, agents have four possible states: Susceptible (S), Exposed (E), Infectious(I) and Removed (R). Only infectious agents can induce the change of state of

another susceptible agent to that of exposed with a given infection rate, $r_{S \to E}$. Each exposed agent necessarily becomes infectious after an incubation time, $t_i$, while each infectious agent remains in this state only during the recovery/removal time interval, $t_r$. In a real epidemic this time would be the interval of time during which an individual remains infectious. The agent can move to the removed compartment either because it dies, recovers or gets isolated. All these outcomes are equivalent as regards the evolution of the epidemic, which is monitored by the total fraction of agents in states $S(t)$, $E(t)$, $I(t)$, $R(t)$ at any given time. The study of an epidemic in terms of these variables is the SEIR paradigm [9]. In this context, the so-called basic reproduction number, $R_0$, is a fundamental quantity that controls the rate of infection in an homogeneous susceptible population. It is defined as the average number of individuals that a given infectious agent turn to exposed in the interval $t_r$, assuming a fully homogeneous and susceptible population.

In this paper we want to derive a set of equations for these global variables that describe correctly the global dynamics under the necessary assumption of a well-mixed and homogeneous population, if seen at a sufficiently large scale, but that incorporates arbitrary distributions of incubation, removal and infection rates in the microdynamics. These set of equations will be presented in the next section, where they will be benchmarked against two types of ABM simulations, that we briefly describe next. In the first type of ABMs, a number $N$ of agents progressing through the S-E-I-R compartments move in an homogeneous space and get exposed by proximity to infected neighbours with some probability. The second type of models assumes the evolution on a network where the agents have a varying number of contacts.

## Spatially homogeneous ABM

We use the MESA package [11] to simulate the spread of an outbreak in a homogeneous population. The agents in the model are called "turtles", following the nomenclature of the NetLogo [12] software. A two-dimensional turtle world is divided in a grid of equal size cells, and inhabited by $N$ turtles. At the initial time the turtles occupy a randomly chosen cell and at each clock tick they do a random move to a neighbouring cells or stay in the same one.

To simulate the evolution of an epidemic, a small number of turtles are infectious, $I_0$, at the initial time, while $S_0 = N - I_0$ are susceptible. At each clock tick infectious (I) turtles can expose any susceptible (S) turtle they find in their neighbourhood. Two turtles are considered neighbours if they share the same cell or are in neighbouring ones.

More concretely, at each time step each susceptible neighbour of an infected turtle, $k$ is exposed with probability $p$. If the neighbour $k$ becomes exposed, the clock time of this event is recorded, $t_{s \to e}^{(k)}$. At each tick all the exposed turtles are examined and eventually turned into infectious, at time $t - t_{s \to e}^{(k)} > t_i$. Again, the time at which the transition to infectious happens is recorded, $t_{e \to i}^{(k)}$, and the turtle remains infectious until it progresses to the recovered compartment of time $t - t_{e \to i}^{(k)} > t_r$.

The basic assumption of homogeneity at large scales or full-mixing of the S-E-I-R populations requires that the probability of infection per contact is small enough. Only in the limit $p \to 0$ we can expect that the infecting agent sees an average population of susceptibles (i.e the turtles have time to diffuse before a second infection succeeds). In this limit, the basic reproductive number is given by

$$R_0 = c \times p \times t_r, \tag{1}$$

where $c$ is the average number of neighbours or contacts per unit time. For the rules described

before, and in the case of a turtle world homogeneous population, $c$ is simply:

$$c = \left(9\frac{N}{A} - 1\right)\epsilon^{-1}, \tag{2}$$

where $A$ is the total number of grid cells in the turtle world and 9 is the number of neighbouring cells of any given cell (including itself). $\epsilon$ is the measure of one time step. Therefore, the infection probability can be obtained from any desired $R_0$ as:

$$p = \frac{R_0}{c \cdot t_r}. \tag{3}$$

The simulation with this $p$ will only match the input value of $R_0$ in the simulation if $p$ is small enough, so that the assumptions that go in the derivation of this formula are satisfied. The simulation is run for a time $t \gg t_r$ and we use a step such that $t_r/\epsilon \sim 30$.

This basic simulation setup has to be supplemented with a prescription to choose the times that turtles remain exposed/infectious. A very common assumption is to take these times exponentially distributed. This can be interpreted as each turtle trying to leave the exposed/infectious compartments at each time tick with a fixed probability, leading to a Poisson process. Other (more realistic) choices of distribution (gamma, Weibull, etc) allow to model some inhomogeneity in the population. In this case, the evolution of the disease is described by a non-Markovian SEIR model.

At each step the software records the turtles in each compartment and thus provides (in the limit of small clock ticks) the functions $S(t)$, $E(t)$, $I(t)$ and $R(t)$, which can be directly compared with the predictions of the solutions of the SEIR models.

## ABM on networks

Realistic populations are not necessarily well-mixed, at least not at small scales. Most individuals have contact only with a very small fraction of the total population. Complex networks show very rich topological features that are similar to real-world social networks. They can have a small number of links between nodes and still display the small-world phenomena. Scale-free networks can also capture the large difference of contacts that different individuals in society have. The study of the evolution of diseases on complex networks allows to study the impact of this rich topological structure in the evolution of and epidemic. The spread of epidemics on networks is an area of intense research. Since the seminal works [13] many studies have been performed on this topic (see the recent review [14] and references therein).

A network is just a non-oriented graph $\{G, E\}$ consisting on nodes $G = \{n_i\}_{i=1}^N$ and edges linking two nodes, $E = \{e_{ij}\}$. We say that two nodes $n_a$ and $n_b$ are connected if $e_{ab} \in E$. The number of edges attached to a node $n_a$ is called its degree and labelled $k_a$.

In the context of the spread of an infectious disease, each node is an agent, and the edges represent the contacts. Each contact links two agents that can expose each other if one is infectious and the other susceptible. The number of edges is therefore the number of contacts. At each tick of time, $\epsilon \ll t_r$, infectious nodes pick a single edge at random, and if susceptible they attempt to infect the node attached to it with probability $p$. In this setup, each click of time represents therefore a contact and all nodes have the same number of contacts per unit time $c = 1$. More general situations can be simulated by allowing infectious nodes to attempt infecting several nodes at each time step. As in the turtle world, the infectious agent remains so in a time interval $t_r$, while the times $t_r$, $t_i$ can be chosen different for each node by drawing samples of some previously chosen distribution.

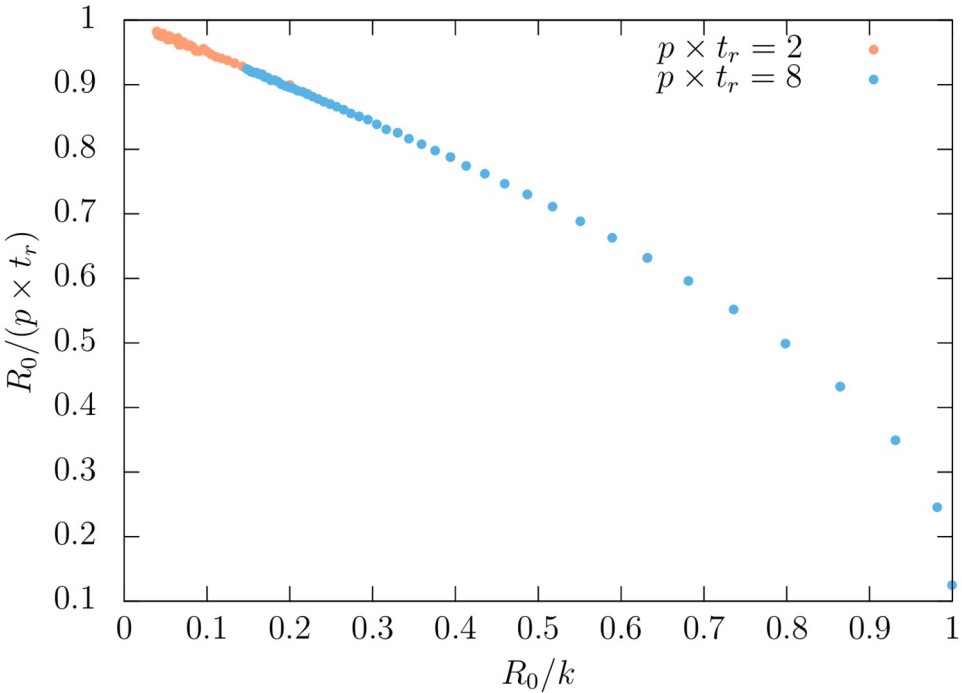

**Fig 1. $R_0/(p \times t_r)$ as a function of $R_0/k$, for a network where each link is randomly linked to $k$ other nodes, for two values of $pt_r = 2, 8$.**

In contrast with the turtle world, a general network breaks the assumption of full mixing, since two nodes that are not linked have exactly zero probability of transmiting the disease between them. On the other hand, a fully connected network, where each node is linked to the remaining $N - 1$ nodes, represents correctly a fully mixed situation. In this case the basic reproductive number $R_0$ is simply

$$R_0 = pt_r. \tag{4}$$

For a general network with small clustering, the correction to this relation is expected to scale as $\propto 1/\langle k \rangle$ as shown in Fig 1). More generally, the value of $R_0$ can depend in a non-trivial way on the network topological properties.

In our study we will concentrate on a particular one-parameter family of random networks described by Klemm and Eguiluz (KE) [15]. These complex networks show a number of features that are expected in realistic networks:

**Scale-free** Nodes with both large and small number of contacts are present. In fact the distribution of the number of nodes is given by the power law

$$P(k) = \frac{\langle k \rangle^2}{2k^3}, \quad k > \frac{\langle k \rangle}{2} . \tag{5}$$

**Small-world** Most nodes are not linked between themselves (i.e. $\langle k \rangle \ll N$), but every link can be reached from any other by a small number of hops. Being more specific, the average distance between nodes gros logarithmically with the size of the network $\langle d \rangle \sim \log N$.

**High clustering** Even if two networks share the number of nodes, edges and the degree distribution, they can look very different if the average clustering coefficient, $\langle C \rangle$, is different.

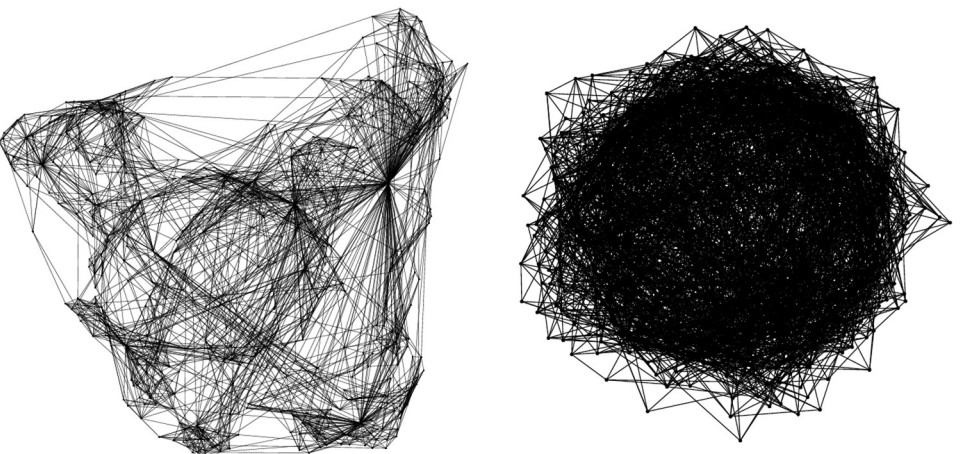

**Fig 2. Two examples of KE network with 500 nodes, mean degree $\langle k \rangle$ = 9.93 and different clustering: $\langle C_i \rangle$ = 0.5 ($\mu$ = 0.1) (left) and $\langle C_i \rangle$ = 0.07 ($\mu$ = 0.9) (right).**

The clustering coefficient $c_i$ of a node $n_i$ measures the probability that two neighbors of $n_i$ are also neighbours

$$C_i \equiv \frac{2|\{e_{jk} \setminus e_{ji}, e_{ki}, e_{jk} \in E\}|}{k_i(k_i - 1)}. \tag{6}$$

In Fig 2 we show two networks with equal distribution of $k$ that differ only in the different clustering properties.

KE networks depend on a free parameter $\mu$ that does not affect the average degree of the network or its distribution, but affects severely the value of the clustering, interpolating from almost no clustering $\langle C \rangle$ = 0 for $\mu \rightarrow 1$, to a very clustered network with $\langle C \rangle \approx 0.84$ for $\mu = 0$.

## uSEIR formulation

A real epidemic is a complex stochastic process that eventually evolves to a regime where there are large numbers of individuals in the S-E-I-R compartments. In the assumption that the populations in these compartements are homogeneous and maximally mixed, the dynamics of the system should be well described in terms of the global variables $S(t)$, $E(t)$, $I(t)$, $R(t)$, whose time evolution is described by a set of deterministic differential equations [9, 16].

We first want to derive the set of equations that should describe the dynamics of these variables under the assumption that the incubation and removal times are fixed. The relation between the changes in these variables is essentially fixed by unitarity. On the one hand, each individual must be in one of the S, E, I or R compartments. Therefore the number of individuals in the population, $N$, is a constant:

$$S(t) + E(t) + I(t) + R(t) = N. \tag{7}$$

Secondly, there must also be a relation between the rates at which these different individuals move from one compartment to the next. An infectious process is that in which an infected individual gets in contact with a susceptible one. Let us call $r_{S \rightarrow E}$ the rate of infection per unit time per infected individual and per susceptible individual. The number of susceptible individuals gets reduced by those that become exposed between $[t, t + dt]$, that is:

$$dS(t) = -r_{S \rightarrow E}I(t)S(t)dt. \tag{8}$$

The parameter $r_{S \to E}$ that governs the infection rate is often denoted as $\beta/N$ in standard SEIR notation. This is the basic equation that assumes an homogenous and maximally-mixed susceptible and infectious populations, making the treatment of the microscopic process in terms of global variables possible. It constitutes the simplest possible form of the force of infection, defined as (minus) the logarithmic derivative of $S$. Keeping within the simplest approximation, if the incubation and removal times of all individuals have the same values, we must also have that the individuals that become exposed at time $t$ are those that move from compartment $S \to E$ minus those that move from $E \to I$. But the latter must be the ones that entered the exposed compartment in time $t - t_i$. Therefore we have:

$$
\begin{aligned}
dE(t) &= -dS(t) + dS(t - t_i)\theta(t - t_i), \\
dI(t) &= -dS(t - t_i)\theta(t - t_i) + dS(t - t_i - t_r)\theta(t - t_i - t_r), \\
dR(t) &= -dS(t - t_i - t_r)\theta(t - t_i - t_r),
\end{aligned}
\tag{9}
$$

where $\theta$ is the Heaviside step function.

The initial conditions to these equations start with a fixed $N$ and a number of infected individuals at time $t = 0$, $I(0) = I_0$, so that $S(0) = S_0 = N - I_0$, while $E(0) = 0$ and $R(0) = 0$. In the equations above, the number of initially infected individuals does not recover, but we can easily force this with the substitution in Eq (8):

$$
I(t) \to \tilde{I}(t) \equiv I(t) - I(0)\theta(t - t_r).
\tag{10}
$$

These equations depend only on three variables, namely $r_{S \to E}$, $t_i$ and $t_r$, which in principle are the same parameters appearing in the classical SEIR models. In terms of the basic reproduction number, $R_0$, $r_{S \to E}$ corresponds to the combination:

$$
r_{S \to E} = \frac{R_0}{N t_r}.
\tag{11}
$$

Note that $R_0$ is proportional to $t_r$, while $r_{S \to E}$ is independent of $t_r$. In a microscopic description of the infected process as in the ABM simulations, the rate is related to the microscopic parameters via $R_0$ from Eq (1) or Eq (4) for the different ABMs.

We can compare the uSEIR and classical SEIR solutions to the ABMs simulations, matching the basic microscopic parameters. In Fig 3 we show the curve for the fraction of infected individuals as a function of time measured from 10 independent turtle simulations in a population of $10^4$ agents with a fraction of infectious agents of $10^{-3}$ at $t = 0$, and assuming fixed parameters $t_i$, $t_r$ and $r_{S \to E}$ for all the agents. The uSEIR solution agrees very well with the simulations, while the classical SEIR predicts a wider and less pronounced peak.

This is of course not surprising, since classical SEIR is known to be valid when $t_i$ and $t_r$ are exponentially distributed, corresponding to an underlying Markovian stochastic process. Modifications of SEIR equations adding more compartments can be designed to represent Erlang distributions, that for sufficiently large $n$ are narrower, but uSEIR gets the limit of fixed values of the parameters directly and without these complications.

In realistic cases, not all individuals have the same incubation or removal times, and certainly not all individuals have the same number of contacts and probability of infection per contact. In the following, we consider the effect of these different non-homogeneities.

## Generically distributed $t_i$ and $t_r$

Non-trivial distributions for $t_i$ and $t_r$ can be incorporated in the uSEIR equations by considering different compartments of individuals. For example, the population divides into those with

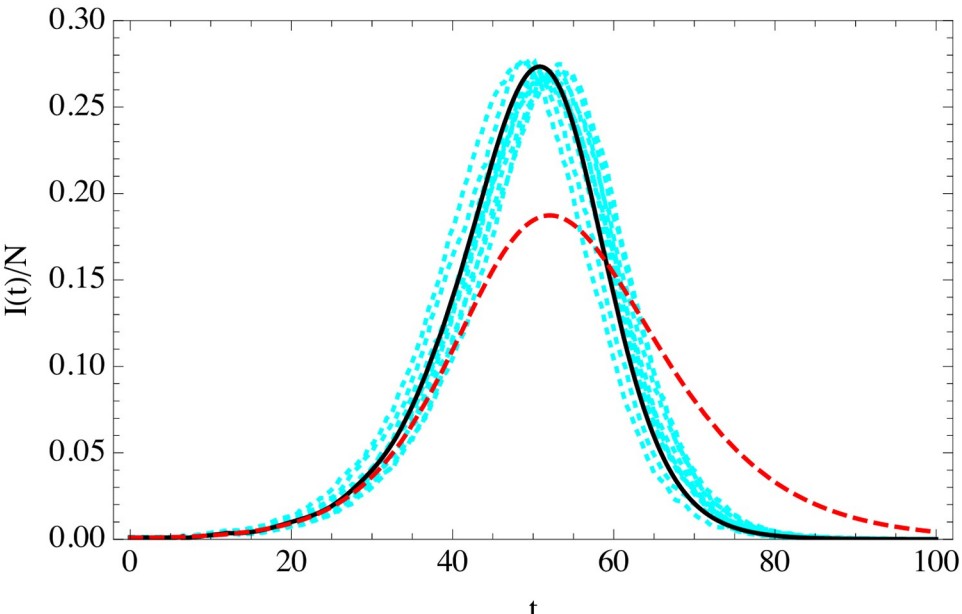

**Fig 3. Curve of the infected individuals as a function of time (in days) for the uSEIR (solid-black), minimal SEIR (dashed-red) and 10 agent simulations (cyan) in a population of $N = 10^4$ and $I(0) = 10$ with $R_0 = 3.5$, $t_i = 5.5$ days and $t_r = 6.5$ days.** The values of $R_0$, $t_i$ and $t_r$ are in the typical range of those used to describe the current COVID-19 pandemic.

different incubation periods, $t_i^{(k)}$, so we have $S_k(t)$ as the susceptible individuals in the $k$-th compartment of incubation time. Each compartment follows its usual progression $S_k \rightarrow E_k \rightarrow I_k \rightarrow R_k$, but the important point to notice is that a given susceptible individual in compartment $k$ becomes an exposed individual in the same compartment $k$, but can get infected from any infectious individual in any other compartment. If we assume that the capability to infect per unit time is independent on the compartment, the number of susceptible individuals in compartment $k$ changes as they become exposed according to:

$$dS_k(t) = -r_{S \rightarrow E} \tilde{I}(t) S_k(t) dt. \tag{12}$$

while Eq (9) will still be valid for the exposed, infected and recovered in each compartment $k$, taking the incubation period as that corresponding to this compartment, $t^{(k)}$.

Summing over all the compartments, the first equation leads to:

$$dS(t) = -r_{S \rightarrow E} \tilde{I}(t) S(t) dt, \tag{13}$$

while in the others we get

$$
\begin{aligned}
dE(t) &= -dS(t) + \sum_k dS(t - t_i^{(k)}) \theta(t - t_i^{(k)}), \\
dI(t) &= \sum_k \{ -dS(t - t_i^{(k)}) \theta(t - t_i^{(k)}) + dS(t - t_i^{(k)} - t_r) \theta(t - t_i^{(k)} - t_r) \}, \\
dR(t) &= \sum_k \{ -dS(t - t_i^{(k)} - t_r) \theta(t - t_i^{(k)} - t_r) \}.
\end{aligned}
\tag{14}
$$

Obviously in the limit of $t_i^{(k)}$ varying continuously the sum becomes an integral with the corresponding PDF, $P_E(t_i)$:

$$\sum_k (...) \rightarrow \int dt_i P_E(t_i)(...), \quad \int_0^\infty dt_i P_E(t_i) = 1. \tag{15}$$

We can similarly assume sub-compartments for varying $t_r$, with a PDF $P_I(t_r)$, and the modification would be analogous, resulting in the following delay integro-differential equations:

$$
\begin{aligned}
\frac{dS(t)}{dt} &= -r_{S \to E} \int dt_r P_I(t_r) \tilde{I}(t) S(t), \\
\frac{dE(t)}{dt} &= -S'(t) + \int_0^t dt_i \; S'(t - t_i) P_E(t_i), \\
\frac{dI(t)}{dt} &= -\int_0^t dt_i \; S'(t - t_i) P_E(t_i) + \int_0^t dt_i \int_0^{t-t_i} dt_r \; S'(t - t_i - t_r) P_E(t_i) P_I(t_r), \\
\frac{dR(t)}{dt} &= -\int_0^t dt_i \int_0^{t-t_i} dt_r \; S'(t - t_i - t_r) P_E(t_i) P_I(t_r).
\end{aligned}
\tag{16}
$$

We refer to Eqs (16) and (9) indistinctively as uSEIR.

A simple and efficient algorithm to solve these equations, complete with easy-to-use codes in Python and Julia, is described in S1 Appendix.

## Recovering classical SEIR

In the case where the probabilities are exponential, the integro-differential equations can be reduced to regular differential ones, of the classical SEIR type.

Let us assume

$$P_E(t_i) = \frac{1}{\langle t_i \rangle} e^{-t_i / \langle t_i \rangle}, \tag{17}$$

and define

$$f(t) \equiv \int_0^t dt_i P_E(t_i) S'(t - t_i) = \int_0^t dz P_E(t - z) S'(z). \tag{18}$$

The derivative of this function is related to that of $E(t)$, using Eq (14),

$$f'(t) = -\frac{1}{\langle t_i \rangle} \frac{dE}{dt}(t), \tag{19}$$

so up to a constant

$$f(t) = -\frac{E(t)}{\langle t_i \rangle} + C. \tag{20}$$

Since $f(0) = E(0) = 0$, the constant must vanish and the equations reduce to:

$$
\begin{aligned}
\frac{dS}{dt} &= -r_{S \to E} \int dt_r P_I(t_r) \tilde{I}(t) S(t), \\
\frac{dE}{dt} &= -\frac{dS}{dt} - \frac{1}{\langle t_i \rangle} E(t), \\
\frac{dI}{dt} &= \frac{1}{\langle t_i \rangle} E(t) - \frac{1}{\langle t_i \rangle} \int_0^t P_I(t_r) E(t - t_r), \\
\frac{dR}{dt} &= \frac{1}{\langle t_i \rangle} \int_0^t P_I(t_r) E(t - t_r).
\end{aligned}
\tag{21}
$$

Analogously we define

$$
g(t) \equiv \frac{1}{\langle t_i \rangle} \int_0^t dt_r P_I(t_r) E(t - t_r),
\tag{22}
$$

which for an exponential with average $\langle t_r \rangle$ satisfies

$$
g'(t) = \frac{I'(t)}{\langle t_r \rangle},
\tag{23}
$$

and therefore

$$
g(t) = \frac{I(t)}{\langle t_r \rangle} + C',
\tag{24}
$$

where $C' = -I(0)/\langle t_r \rangle$. Finally, the integral in the first equation:

$$
\int dt_r P_I(t_r) \tilde{I}(t) = I(t) - I(0)(1 - e^{-t/\langle t_r \rangle}) \equiv \bar{I}(t).
\tag{25}
$$

Finally, defining

$$
\bar{R}(t) \equiv R(t) + I(0)(1 - e^{-t/\langle t_r \rangle}),
\tag{26}
$$

we recover the classical SEIR equations:

$$
\begin{aligned}
\frac{dS}{dt} &= -r_{S \to E} \bar{I}(t) S(t), \\
\frac{dE}{dt} &= -\frac{dS}{dt} - \frac{1}{\langle t_i \rangle} E(t), \\
\frac{d\bar{I}}{dt} &= \frac{1}{\langle t_i \rangle} E(t) - \frac{1}{\langle t_r \rangle} \bar{I}(t), \\
\frac{d\bar{R}}{dt} &= \frac{1}{\langle t_r \rangle} \bar{I}(t).
\end{aligned}
\tag{27}
$$

We can incorporate easily the exponential distributions for $t_i$ and $t_r$ in the ABMs simulations, while we maintain the rate of infection constant. The comparison of the SEIR solution of Eq (27) and the homogeneous ABM simulation with exponentially distributed $t_i$ and $t_r$ is shown in Fig 4. The agreement as expected is good, even if the variance is much larger than in the fixed-parameters case. In fact, an interesting observation is that most of the observed variance of the outbreaks is a simple time translation. If we time-translate all the outbreaks to

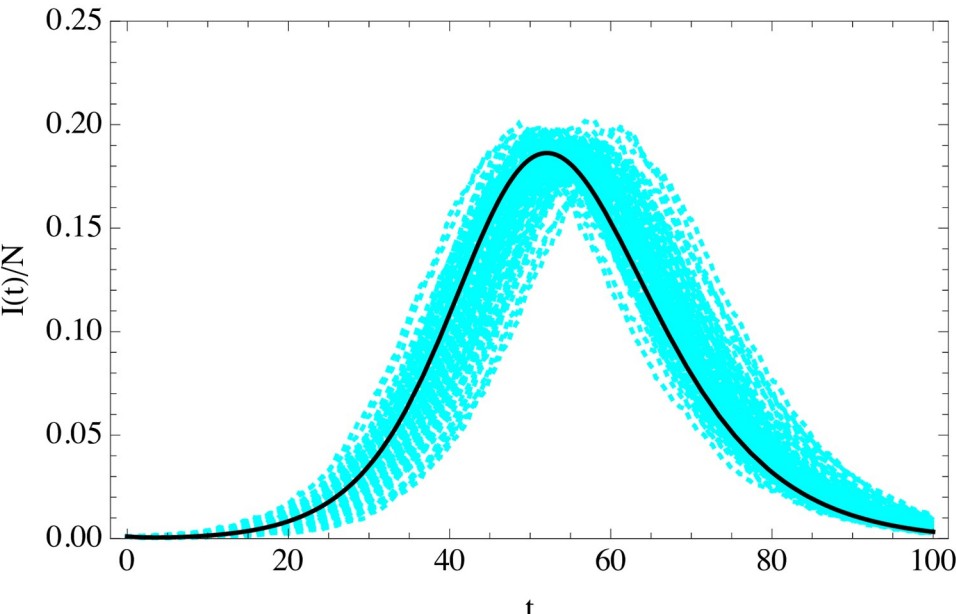

**Fig 4. Evolution of the fraction of infected individuals as a function of time (in days) in classical SEIR (solid black) of Eq (27), and in 100 random turtle simulations with exponentially-distributed $t_i$ and $t_r$ (cyan) in a population of $N = 10^4$ and $I(0) = 10$ with $R_0 = 3.5$, $\langle t_i \rangle = 5.5$ days and $\langle t_r \rangle = 6.5$ days.**

make their maxima coincide the variance is much smaller and the agreement with SEIR better, see Fig 5. We will discuss the origin of this time-shift in the following section.

An exponential distribution for the incubation and removal times is however not realistic. A more realistic distribution seems to be, e.g., a general gamma distribution, $\Gamma[k, \theta]$. For

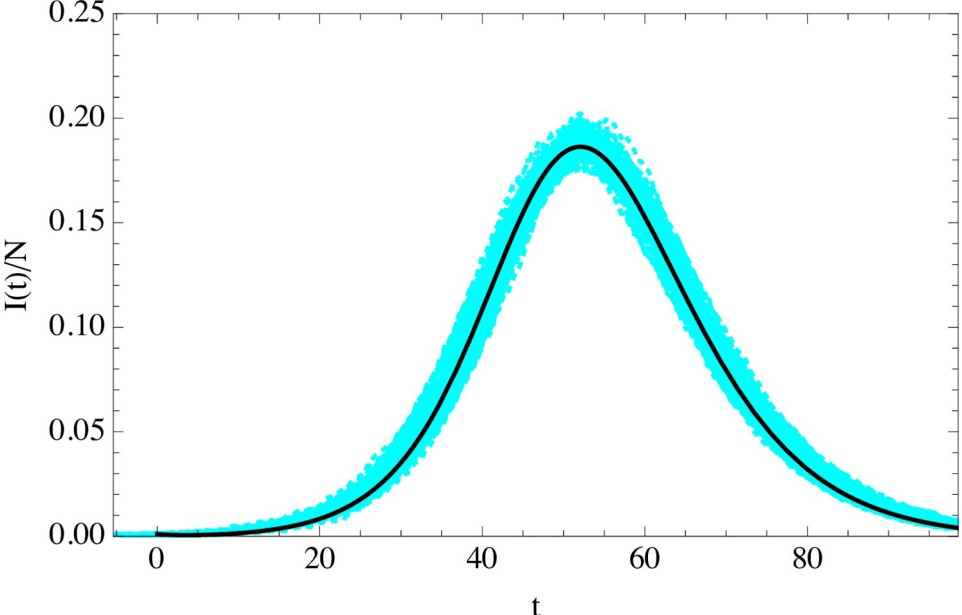

**Fig 5. As in Fig 4, after a time-shift of the simulation curves so that their maxima coincide with that of the SEIR solution.**

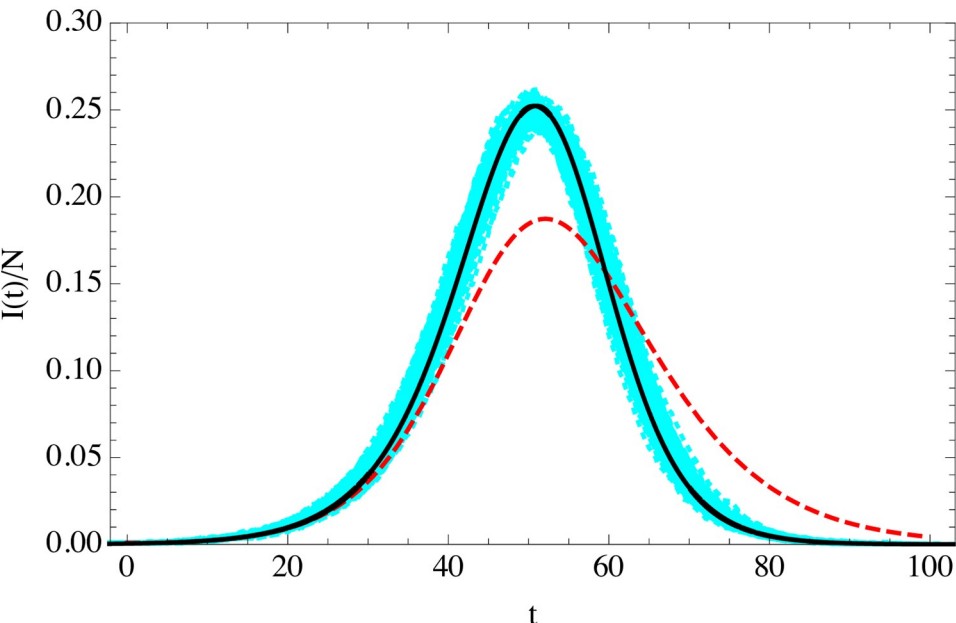

**Fig 6. Fraction of infected individuals as a function of time (in days) from the average of 100 turtle simulations with $t_i$ and $t_r$ distributed in the population according to the gamma distributions (cyan) compared to the solution of the uSEIR of Eq (16) (solid black) and classical SEIR (red dashed).** The simulations have been time-shifted so that their maxima coincide. The simulation has parameters in both cases $N = 10^4$ and $I(0) = 10$ with $R_0 = 3.5$, $\langle t_i \rangle = 5.5$ days and $\langle t_r \rangle = 6.5$ days.

the COVID-19 epidemic, the distribution of incubation times has been shown to be well described with a gamma with parameters $(k, \theta) \simeq (5.8, 0.948)$ [7], corresponding to an average $\langle t_i \rangle \simeq 5.5$ days. For the removal time, we assume the same distribution with parameters $(6.5, 1)$. This corresponds to the average between the "short" and "long" removal times discussed in [7].

We compare the results of the gamma-distributed ABM simulations in Fig 6. As expected, classical SEIR does not give a good description of the simulations in this case, while solving the integro-differential Eq (14) does.

We note that, although there is a vast literature on incorporating arbitrary distributions for $t_i$ and $t_r$ in stochastic approaches to the propagation of epidemics (see, e.g., a recent review in [8]), we have not found a simple formulation of the problem in the context of compartmental models such as the one described by equations Eq (16). While generalizations of exponential distributions, such as the Erlang case, are dealt with in the standard SEIR literature using a superficially similar sub-compartmentation approach (see, e.g., [17–21]) this is not quite as general as the treatment described here.

## Non-uniform infection rate and universality

A different situation is when the rate of infection is non-uniform across the population. It is important to stress that the rate depends on two independent parameters: the number of contacts per infected individual, which critically depends on the clustering properties of the social network, and the probability of infection per contact. Non-uniformity can originate in either of the two properties. In this section we will consider the simplest case of a uniform number of contacts, but a non-uniform infection probability per contact.

## Non-uniform rate: The probability of infection

We could separate the population in individuals that infect others with different rates. The rate might depend on the type of infectious individual and the type of susceptible individual. Defining $r_k^l$ to be the rate at which an infected individual of type $l$ infects a susceptible individual of type $k$. The equations in this case are:

$$
\begin{aligned}
dS_k(t) &= -\sum_l r_k^l I_l(t) S_k(t) dt, \\
dE_k(t) &= -dS_k(t) + dS_k(t - t_i)\theta(t - t_i), \\
dI_k(t) &= -dS_k(t - t_i)\theta(t - t_i) + dS_k(t - t_i - t_r)\theta(t - t_i - t_r), \\
dR_k(t) &= -dS_k(t - t_i - t_r)\theta(t - t_i - t_r).
\end{aligned}
$$

where $t_i$ and $t_r$ might also depend on the compartment.

Assuming that the rates only depend on the type of infecting individual and not on the type of susceptible and, for simplicity, that $t_i$ and $t_r$ are fixed, only the total number of individuals in each compartment needs to be evolved. This is the case, because the different compartments are in some proportion in the population and we assume the proportion is preserved by the initial conditions of the $I_k(0)$ and $S_k(0)$. The equations reduce to the usual ones with a rate that is the weighted average:

$$
r_{\text{eff}} = \sum_k r^k p^k, \tag{28}
$$

where $p^k$ is the proportion of individuals in compartment $k$. In the continuous case

$$
r_{\text{eff}} = \int dr \; r P_R(r), \tag{29}
$$

where $P_R(r)$ is the corresponding PDF.

However, this result seems in conflict with the fact non-uniformity in the rate is known to be very important in the evolution of an epidemic (see, e.g., [22, 23]). One example of this is the relevance of the fraction of individuals for which the probability of infection is zero. Their presence in a given population implies that the effective number of useful contacts gets reduced. When the fraction of the population with zero infecting power is large enough the epidemic may be aborted. In practice, the effect is similar to that of herd immunity, used to measure the needed number of vaccinations to abort an epidemic. A very rough estimate for the fraction of herd immunity, $f_H$, would be

$$
R_0(1 - f_H) = 1, \;\; f_H = 1 - 1/R_0. \tag{30}
$$

For example, with $R_0 \sim 3, f_H \sim 0.7$, that is 70% of the population. One would then naively expect that in an epidemic where this estimate holds about 70% of the population ends up getting infected; however, in the previous examples a larger fraction is found. The reason for this overshooting effect is that, due to the time delay in the process, the fraction of recovered individuals grows slowly and is not effective in reducing the growth of the epidemic sufficiently, as would be the case if the fraction of immune individuals had been present from the start, as would be the case, for instance, in a (partially) vaccinated population. Note that in the SEIR paradigm, the immune population is part of the susceptible, that pass by the compartments $E \rightarrow I \rightarrow R$ but have zero infecting power when they are $I$ so they are inert. In practice the evolution of the epidemic would be identical if we just dropped them from the start and readjust the rate not to include them.

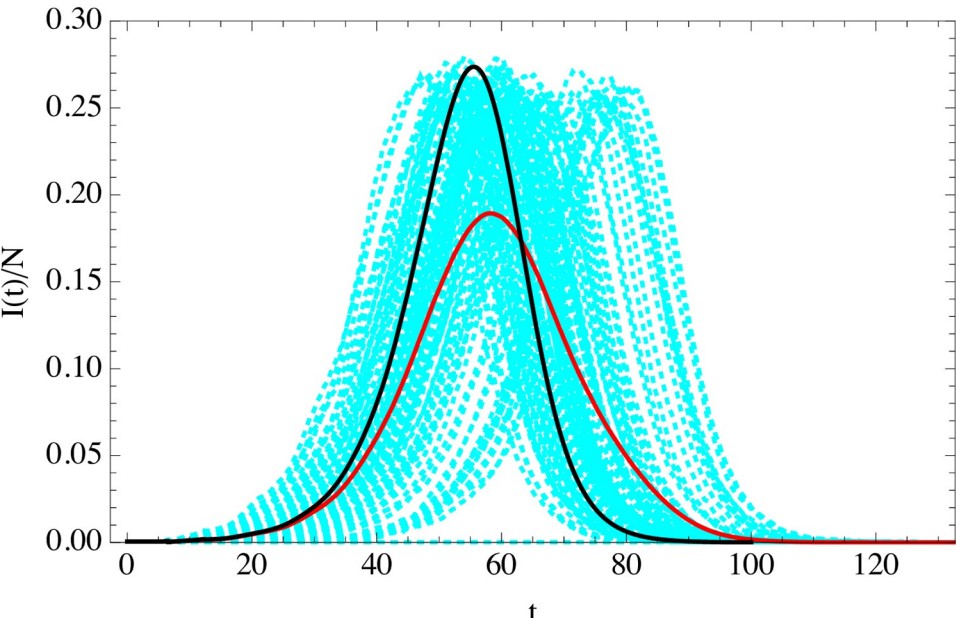

**Fig 7. Curve of the fraction of infected individuals as a function time (in days) from the average of 100 agent simulations with $R_0$ distributed in the population according to negative binomial (cyan) with $t_i$ and $t_r$ fixed.** The average of those histories is the red curve. The simulation has parameters $N = 2 \times 10^4$ and $I(0) = 10$ with $\langle R_0 \rangle = 3.5$, $t_i = 5.5$ days and $t_r = 6.5$ days. This is compared to uSEIR (black).

It has been argued that for COVID-19 the distribution of $R_0$ across the population is well described by the negative binomial distribution, NB[0.16, 0.0437] [24], which has average 3.5 but a large dispersion. This distribution implies that about 60% of the population is immune (not far from the naive herd immunity), while there must be few individuals that have a very large rate of infection, the famous superspreaders.

In Fig 7 shows the evolution of 100 simulations assuming fixed $t_i$ and $t_r$ while $R_0$ is drawn from this negative binomial. The average of those outbreaks as well as the result of uSEIR using the average $\langle R_0 \rangle$ are also shown. Clearly the variance is huge, and the average is not a good representation of the individual epidemic histories. The uSEIR curve misses completely the outliers.

There is an interesting observation however. If all the curves are time-translated to make their maxima coincide, they fall in the uSEIR curve, as shown on the right Fig 8.

This fact can be interpreted as follows. The position of the peak is non-universal, because it depends very sensitively on the initial conditions, in particular on what is the infectious potential of the first infectious agents. Since all epidemics start with a small number of individuals, we cannot invoke the central limit theorem for the initial stages of an outbreak. These stages have a large variability, however as the exponential grows the averaging effect of the population starts to be effective. The curve around the maximum is in fact universal, in the sense that it depends on the average of the basic parameters and not on the initial conditions, as we now show from the uSEIR equations.

## Universality and the logistic curve

We have observed that the main effect of the different initial conditions is a temporal shift of the maximum, but the shape or the height of the infection curve does not change significantly. This strongly suggest that the equations have a universal solution. We have indeed found it.

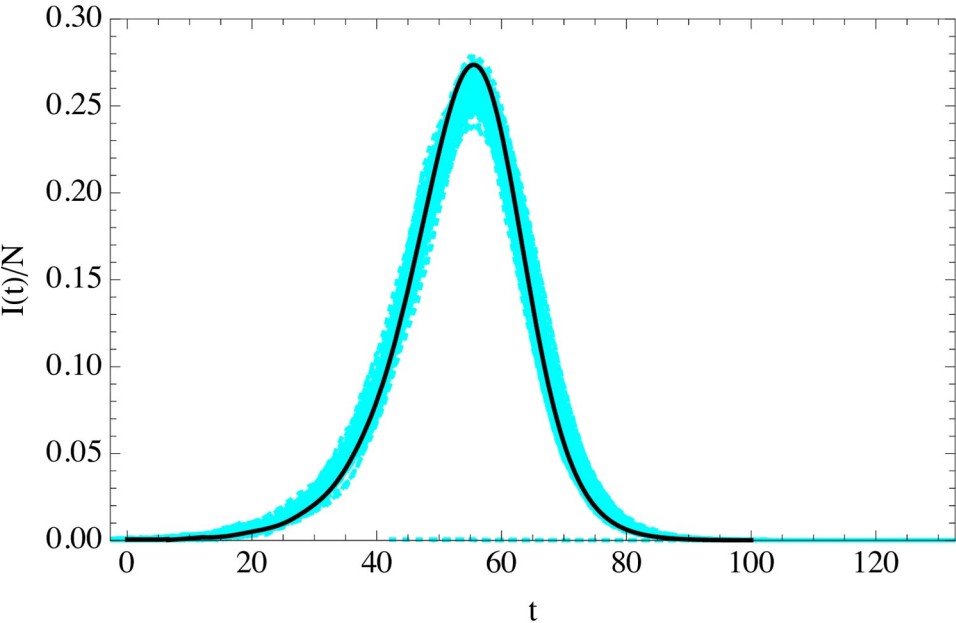

**Fig 8. The same as in Fig 7 with the individual ABM simulations time-shifted to keep their maxima invariant and coinciding with maximum of the uSEIR curve.**

Let us consider the differential Eq (9) near the maximum of the infection curve $t_{\max}$, which will remain as a free parameter. Let us also assume that $t_{\max} \gg t_i, t_r$, and define the function

$$F(t) \equiv S(t)I(t). \tag{31}$$

The differential equations for the uSEIR with fixed $t_i$ and $t_r$ and for $t \gg t_i, t_r$:

$$
\begin{aligned}
\frac{dS}{dt} + \frac{dR}{dt} &= rF(t - t_i - t_r) - rF(t) \simeq -r(t_i + t_r)\left(F'(t) - \frac{t_i + t_r}{2}F''(t)\right), \\
\frac{dE}{dt} &= r(F(t) - F(t - t_i)) \simeq rt_i\left(F'(t) - \frac{t_i}{2}F''(t)\right), \\
\frac{dI}{dt} &= r(F(t - t_i) - F(t - t_i - t_r)) \simeq rt_r\left(F'(t) - \left(t_i + \frac{t_r}{2}\right)F''(t)\right).
\end{aligned}
\tag{32}
$$

which implies

$$
\begin{aligned}
S(t) + R(t) &= C - r(t_i + t_r)\left(F(t) - \frac{t_i + t_r}{2}F'(t)\right), \\
E(t) &= C' + rt_i\left(F(t) - \frac{t_i}{2}F'(t)\right), \\
I(t) &= C'' + rt_r\left(F(t) - \left(t_i + \frac{t_r}{2}\right)F'(t)\right).
\end{aligned}
\tag{33}
$$

Since $I(t) \to 0$, $E(t) \to 0$, $F(t) = S(t)I(t) \to 0$ as $t \to \infty$, it follows that $C' = 0$, $C'' = 0$ and $C = N$. Using the previous equations, it is easy to derive a differential equation for $F(t)$, expanding at

linear order in $t_i$ and $t_r$:

$$F''(t) - \frac{F'(t)^2}{F(t)} + r^2 \frac{t_r}{t_i + \frac{t_r}{2}} F(t)^2 = 0. \tag{34}$$

We are interested in the solution near the maximum, so we use the initial conditions:

$$F'(t_{\max}) = 0, \quad F(t_{\max}) = F_0. \tag{35}$$

This non-linear equation has an analytical solution given by:

$$F(t) = F_0(1 - \tanh^2[a(t - t_{\max})]), \tag{36}$$

with

$$a \equiv r\sqrt{\frac{t_r F_0}{2t_i + t_r}}. \tag{37}$$

This is the universal function that drives the evolution of the infected, exposed and suscepti-ble+recovered individuals near the maximum. The maximum of the infected is at $t_{\max} - t_i$ for the infected, while the maximum(minimum) for the exposed (susceptible+recovered) is at $t_{\max}$. The integral of this function from $[-\infty, \infty]$ is

$$\int_{-\infty}^{\infty} dt F(t) = \frac{2F_0}{a}. \tag{38}$$

Note that for large $t_{\max}$, the range $t < 0$ gives a negligible contribution. We can also derive the value of the susceptible at $t_{\max}$ since

$$S(t_{\max}) = \frac{F(t_{\max})}{I(t_{\max})} = \frac{1}{rt_r}, \tag{39}$$

and the curve of the susceptible can be easily obtained

$$S(t) = S(t_{\max}) - r \int_{t_{\max}}^{t} F(t). \tag{40}$$

The total number of susceptible at the end of the epidemic is therefore:

$$S(\infty) = \frac{1}{rt_r} - r\frac{F_0}{a}. \tag{41}$$

With this we conclude that the epidemic curve is universal once the value of the maximum position is determined. The value of $F_0$ should also depend on the basic parameters and not the initial conditions, although the precise value is not easy to get. A rough estimate can be obtained as follows. Near the maximum, and if the incubation and removal times are suffi-ciently small, we can approximate that $R(t_{\max}) \simeq I(t_{\max}) + E(t_{\max})$, since the infected and exposed quickly recover; using this and the value of $S(t_{\max})$ we can estimate $F_0$ to be

$$F_0 \sim \frac{N - S(t_{\max})}{2r(t_r + t_i)}. \tag{42}$$

The only dependence on the initial condition remains in $t_{\max}$. In Fig 9 we compare the numerical solution to the uSEIR equations to the analytic expression of Eq (36), fixing the parameters $F_0$ and $t_{\max}$ (the height and the position of the peak) from the numerical solution. Varying the initial conditions, that is the fraction of the number of infected individuals at $t = 0$,

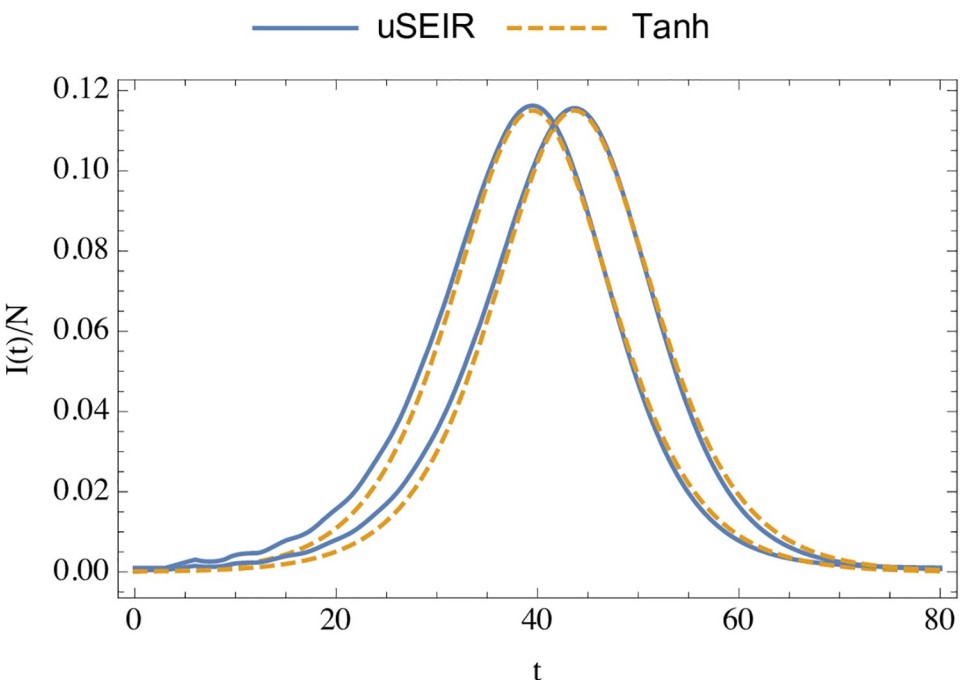

**Fig 9. Comparison of the results of the curve of infected as a function of time (in days) for fixed parameters $R_0$ = 2.1, $t_i$ = 3 days and $t_r$ = 3 days, and the analytical result of Eq (36) with the parameters $F_0$ and $t_{max}$ tuned with the height and position of the peak.** The two pairs of solid curves correspond to a fraction of infected individuals of $10^{-3}$ and $5 \cdot 10^{-4}$. The two dashed lines are the same function shifted in time.

shifts $t_{max}$, but otherwise leaves the curve invariant. As can be seen, the analytical solution around the peak describes very well the full uSEIR solution. The agreement is better for smaller values of $t_i$ and $t_r$.

It is possible to extend this asymptotic solution to the case where $t_i$ and $t_r$ are not fixed but drown from distributions $P_E(t_i)$ and $P_I(t_r)$. Eq (34) gets modified in that the coefficient of the last term becomes:

$$r^2 \frac{\langle t_r \rangle}{\langle t_i \rangle + \frac{\langle t_r^2 \rangle}{2\langle t_r \rangle}}, \tag{43}$$

where $\langle \rangle$ refers to the average with the corresponding PDF. Therefore the logistic, Eq (36), is still the asymptotic solution with a modified parameter:

$$a \rightarrow r\sqrt{\frac{\langle t_r \rangle F_0}{2\langle t_i \rangle + \langle t_r^2 \rangle / \langle t_r \rangle}}. \tag{44}$$

## Non-uniform rate in network simulations

We now consider the non-homogeneities in the social contacts. We have generated a number of KE networks with $\langle k \rangle$ = 40 and different clustering properties, by changing the $\mu$ parameter. The networks have $10^6$ nodes. On these networks we evolve the epidemic using the time progression explained above, starting with 10 infected nodes. The probability of infection per contact is $p = 2 \times 10^{-3}$. For the incubation and removal times we assume a LogNormal distributions, in units of the step time, $\epsilon$, with parameters $(\mu_X, \sigma_X) = (10^3, 200)$ for $t_r$ and $(\mu_X, \sigma_X) = (500, 100)$ for

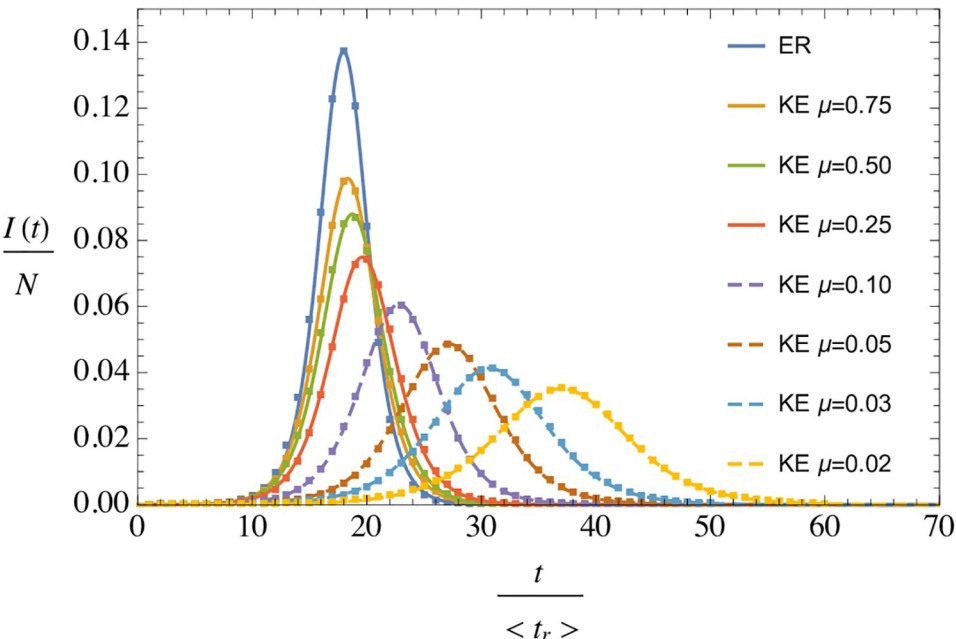

**Fig 10. Average fraction of infected nodes as a function of time in units of $\langle t_r \rangle$ for various networks with equal average degree, $\langle k \rangle = 40$, but different clustering properties, depending on $\mu$.** The ER network shows the result on an Erdős-Rényi random network [25] with $\langle k \rangle = 40$ and zero clustering. The lines going through the data are fits to Eq (36), leaving $a$, $I_0$ and $t_{max}$ as free parameters.

$t_i$, where $\mu_X$ is the mean and $\sigma_X^2$ is the variance. For these parameters, $p\langle t_r \rangle = 2$, which gives an approximation to $R_0$ up to $1/\langle k \rangle$ corrections, as explained in sec. 1. From Fig 1, we can get a more precise estimate of $R_0 = $ . For each network we run a number of simulations and average the S-E-I-R fractions, after performing a time-shift to make their maxima coincide (which as in previous cases, reduces most of the variance). In Fig 10 we show the evolution of infected individuals as a function of time for the various networks. We observe a clear dependence on the clustering parameter, but nevertheless the data in all cases is extremely well described by the universal behaviour derived from uSEIR, Eq (36). The lines are three-parameter fits ($a$, $I_0$, $t_{max}$) of the form:

$$I(t) = I_0[1 - \tanh^2(a(t - t_{max}))]. \tag{45}$$

uSEIR predicts, according to Eqs (36) and (44) and Fig 1,

$$\frac{a\langle t_r \rangle}{\sqrt{I_0/N}} = \sqrt{\frac{R_0\langle t_r \rangle}{(2\langle t_i \rangle + \langle t_r^2 \rangle/\langle t_r \rangle)}} \simeq 0.972, \tag{46}$$

while

$$I_0 = p\langle t_r \rangle \frac{F_0}{N}, \tag{47}$$

and using the rough estimate of Eq (42), we get $I_0/N \sim 1/6$. Both parameters would therefore be given in terms of the average microscopic parameters.

In Fig 11 we show the dependence of $a\sqrt{I_0/N}^{-1}$ and $I_0/N$, on the average local clustering $\langle C \rangle$. For small clustering we observe that $a\sqrt{I_0/N}^{-1}$ is roughly constant and matches rather

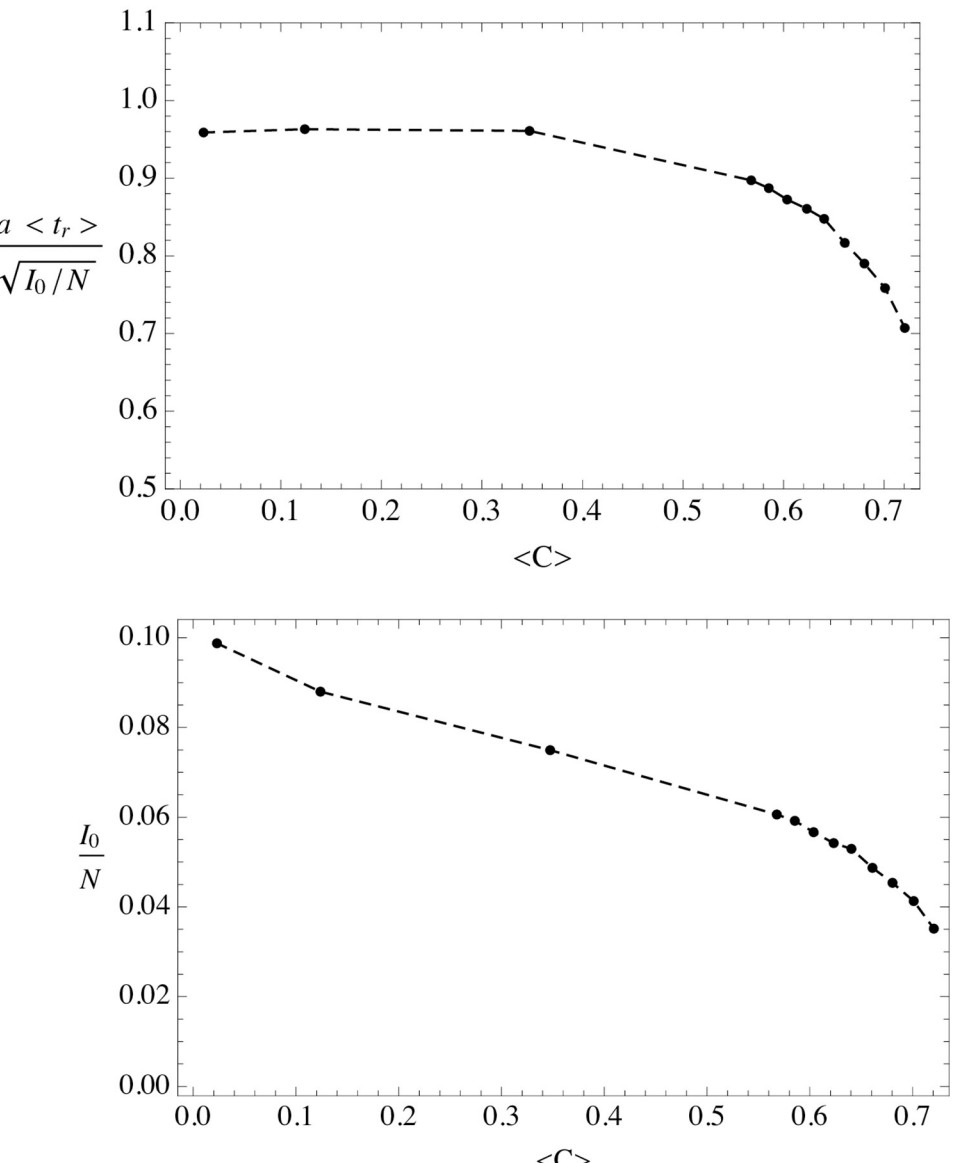

**Fig 11. Dependence of the fit parameters $a\langle t_r\rangle\sqrt{I_0/N}^{-1}$ and $I_0/N$ on the average clustering.** Dashed lines are intended to guide the eye through the data points.

well the microdynamical average value of Eq (37). Instead $I_0/N$ decreases with clustering, even for small clustering. This effect can be interpreted as effective suppression of the fraction of susceptible population: clustering seems to screen the access to the susceptible. Note that if we substitute in the uSEIR equations $S$ by $f_c S$, where $f_c$ is the screening factor, the asymptotic solution is as in Eq (45) with $I_0 \rightarrow f_c I_0$, while $a\sqrt{I_0/N}^{-1}$ remains invariant. This could explain the behaviour found at small clustering.

At large clustering, on the other hand, the parameters $I_0$ and $a$ show a non-trivial dependence with clustering. In spite of this, the logistic remains an extremely good description of the time evolution of the infected fraction. It would be interesting to understand this behaviour in

terms of a renormalization or screening of the basic parameters, or modifications of the force of infection with respect to the well-mixed approximation yielding $-S'/S \propto I$.

As a final comment, we note that everything we have studied here assumes no time variation of the basic parameters. In a real epidemic, measures of social distancing, self-protection, etc. are taken, that induce a sudden change of the basic parameters, particularly the rate of infection. This effect induces a quench of the epidemic curves that we have been discussing in this paper. It will be interesting to explore to what extent the evolution after the quenches can be understood in terms of the fundamental parameters, in particular whether the universality near the herd-peak translates into some universality of the curve after a quench, if it has happened in the asymptotic regime.

## Conclusions

In this paper we have presented a simple formulation, uSEIR, Eq (16) of the SEIR modelization of a epidemic outbreak that properly accounts for an arbitrary distribution of incubation and removal times, reducing to classical SEIR in the limit of distributions of the exponential family. We have compared this model with a series of ABM homogeneous simulations for various scenarios including fixed values for the incubation and removal times, as well as various realistic distributions for the the latter, or for the probability of infection per contact. We have also considered ABM simulations on scale-free networks with varying clustering properties. In all cases, the model reproduced the simulations accurately after a non-universal time-shift. Only in the presence of large local clustering in the distribution of contacts we observed a clear deviation, when the averages of microdynamical parameters are included. The uSEIR formulation allowed us to understand the universality property observed in different outbreaks in the simulations. This derives from an explicit asymptotic solution found for small incubation and removal times in terms of a logistic curve, with a shape that can be determined in terms of the microdynamical parameters. This curve is found to fit very well the data even in cases of large clustering, provided the parameters are left as free fit parameters, suggesting that the dynamics in the high clustering regime may still be well described in terms of global variables but with screened or renormalized parameters. On the contrary, the early stages of an outbreak are highly non universal, an aspect that should be carefully taken into account when fitting data and predicting using any SEIR modelling. Only when the early stages of exponential growth are well underway, is uSEIR expected to be a good description. The averaging of independent outbreaks, without taking into account the non-universal time-shift, can also be very misleading.

## Supporting information

**S1 Appendix.**
(PDF)

## Acknowledgments

We thank J. Hernando for useful discussions.

## Author Contributions

**Conceptualization:** Pilar Hernández, Carlos Pena, Alberto Ramos, Juan José Gómez-Cadenas.

**Formal analysis:** Pilar Hernández, Carlos Pena, Alberto Ramos.

**Methodology:** Pilar Hernández.

**Software:** Alberto Ramos, Juan José Gómez-Cadenas.

**Writing – original draft:** Pilar Hernández.

**Writing – review & editing:** Carlos Pena, Alberto Ramos, Juan José Gómez-Cadenas.

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
