## [Decision Letter · Decision Letter 0]

21 Sep 2020

PONE-D-20-15952

A new formulation of compartmental epidemic modelling for arbitrary distributions of incubation and removal times

PLOS ONE

Dear Dr. Hernández,

Thank you for submitting your manuscript to PLOS ONE. After careful consideration, we have decided that your manuscript does not meet our criteria for publication and must therefore be rejected.

I am sorry that we cannot be more positive on this occasion, but hope that you appreciate the reasons for this decision.

Yours sincerely,

Ram K. Raghavan

Academic Editor

PLOS ONE

Additional Editor Comments (if provided):

Dear Dr. Hernández, Your manuscript was reviewed by two experts in the field, in addition to myself, and I regret to inform that it is not acceptable for publication in PLoS ONE in its current state. Although Reviewer 1 has given favourable comments, Reviewer 2 has raised many valid concerns that I happen to fully agree with. I hope these comments will be useful when you reconsider this work for publication elsewhere.

Reviewers' comments:

Reviewer's Responses to Questions

**Comments to the Author**

1. Is the manuscript technically sound, and do the data support the conclusions?

Reviewer #1: Yes

Reviewer #2: Yes

2. Has the statistical analysis been performed appropriately and rigorously? 

Reviewer #1: I Don't Know

Reviewer #2: Yes

3. Have the authors made all data underlying the findings in their manuscript fully available?

Reviewer #1: Yes

Reviewer #2: Yes

4. Is the manuscript presented in an intelligible fashion and written in standard English?

Reviewer #1: Yes

Reviewer #2: Yes

5. Review Comments to the Author

Reviewer #1: This is an interesting and I think quite useful approach to a long-lamented issue in compartmental models for infectious diseases, that of the non-exponentially distributed transition times. I appreciate the work that has gone into the approach.

My only large comment is on the use of the word uniform. I originally misinterpreted that to mean uniformly distributed, when I believe you meant homogenous or fixed. I would recommend changing the wording.

Specific comments:

Pg 1: SEIR is susceptible-exposed-infectious-removed

Pg 3: It was not clear whether this ABM used fixed values for t_i or t_r

Pg 16: The last line appears to be missing some text

Reviewer #2: The authors present a formulation of compartmental modelling to include arbitrary distributions of incubation and removal times. While the authors claim that the conventional paradigm for compartmental epidemiological modelling is the SEIR model, and that this model assumes exponentially distributed incubation and removal times. A minor point: SEIR is not the conventional paradigm for compartmental epidemiological modelling. It is one type of compartmental model structure, but its appropriateness depends on 1) the pathogen being studied, 2) the research question, and 3) data availability. There are many different model structures (SI, SIS, SIR, SEIR) or adaptions thereof (SIIBS, SEIRS, SEIHFR), and individuals can have different states than just S, E, I, and R, although it is recognized to be an appropriate model choice for Covid-19.

Nonetheless, the assumption that incubation and other transition times are exponentially distributed is indeed well known. However, there already exist various types of distribution delay models and renewal equations, which can cover all alternatively assumed distributions, in a similar fashion as the authors propose in this paper. Even the 1927 Kermack and McKendrick model allows for this. They generally all require the introduction of additional strata in the model, the capability to track the time at which individuals were infected, and some sort of integro-differential equation to solve them (see e.g. Breda et al 2012 On the formulation of epidemic models (an appraisal of Kermack and Mckendrick)). Although they are more realistic, this comes at the added cost of computational inefficiency, which can be a real bottleneck when hundreds of thousands of models need be ran to e.g. fit the model to observed data. It is this reason why many modellers opt for a sequence of e.g. different infectious compartments to implement an Erlang rather than exponential distribution, as a better trade-off between realism and computational efficiency. Although this is not the case for the initially presented uSEIR model with homogeneous values for ti and tr for all individuals in the population, this model formulation is probably even more unrealistic than the assumption regarding exponential distributions, so there would always be a need to incorporate multiple strata in the model.

There are several other, smaller points:

- The authors refer to agents in their ABM model as 'turtles' to stick with the nomenclature of this specific software package, which is fine, but it would be good to stick to other more basic epidemiological notation as well to improve readability of the manuscript for individuals with an epidemiological background, which I assume is the target audience for this paper. E.g., the effective contact rate (which multiplied with I(t) is the force of infection) is referred to as the infection rate, and denoted as "Rs->e" rather than the more conventional symbols using c, Beta or Lambda. When a susceptible individual is infected, the authors refer to this as being exposed (the Exposed compartment in the SEIR is an unfortunate standard naming convention, and is better referred to as Preinfectious).

- The authors find "an interesting observation is that most of the observed variance of the outbreaks is a simple time translation". This is a well known phenomenon as the stochastic effects of an epidemic are especially important when modelling 1) an early outbreak or 2) a small population. Related, on page 14, authors have observed that, with a given R0, the peak and shape of the epidemic does not differ substantially. For immunizing infections, it is indeed a well known fact, as the epidemic will peak once the herd immunity threshold is reached, or stated otherwise, when the proportion of the population that is susceptible is equal to 1 /R0.

- The usual terminology for the universal equation starting at page 14 is the 'final size equation'. On page 12, the authors observe that the total number of infections exceed the herd-immunity threshold. This is another well known phenomena and is usually referred to as an epidemic 'overshooting'.

- What does theta in equation 9 refer to?

- Authors give estimates for R0 for Covid-19 of 3 and 3.5. As R0 values are extremely context specific, it would be good to state in which populations these values have been estimated.

- On page 12, authors refer to "individuals for which the probability of infection is zero,which in practice makes them immune". They are here referring to infected individuals, which by definition means they are not immune, so I would at least use a different term for these individuals. Related to this, the authors claim that due to the estimated negative binomial distribution of R0, 60% of the population must be immune. I see how the 60% value is calculated from the CDF, but do not understand this interpretation. Yes, many individuals will have a low number of contacts and will therefore only infect few (or some 0) secondary cases, but this does not mean that they are immune, which would imply they would not have been infected themselves. In fact, the ones resulting in few secondary cases are probably the very young or very old, which are the age-groups at highest risk for severe disease and death, so they would leave a very real mark on the epidemic progression once other outcomes than just infections are incorporated.

6. PLOS authors have the option to publish the peer review history of their article (what does this mean?). If published, this will include your full peer review and any attached files.

Reviewer #1: No

Reviewer #2: No

- - - - -

---

## [Author Response · Author response to Decision Letter 0]

17 Oct 2020

This information has been added in a filed AsnwersToReferees in the provided material.

---

## [Editor Report · Decision Letter 1]

18 Nov 2020

PONE-D-20-15952R1

A new formulation of compartmental epidemic modelling for arbitrary distributions of incubation and removal times

PLOS ONE

Dear Dr. Hernández,

Thank you for submitting your manuscript to PLOS ONE. After careful consideration, we feel that it has merit but does not fully meet PLOS ONE’s publication criteria as it currently stands. Therefore, we invite you to submit a revised version of the manuscript that addresses the points raised during the review process.

While the work meets the criteria of PLOS ONE, you should provide an improved introduction that places your work into historical context. Reviewer 2 provided a guide for this context by pointing out the generalizability of Kermack and McKendrick's work as well as the associated difficulties. Also, your notation uSEIR seems arbitrary - does the "u" stand for something. If so, please describe. If not, it would be worthwhile to use meaningful nomenclature.

We look forward to receiving your revised manuscript.

Kind regards,

Eric Forgoston

Academic Editor

PLOS ONE

Journal Requirements:

2. Please clarify in your Data availability statement what can be found at each link. Were any datasets used or generated in the study, and if so, are these publicly available?

3. We note that your manuscript is not formatted using one of PLOS ONE’s accepted file types. Please reattach your manuscript as one of the following file types: .doc, .docx, .rtf, or .tex (accompanied by a .pdf).

If your submission was prepared in LaTex, please submit your manuscript file in PDF format and attach your .tex file as “other.”

---

## [Author Response · Author response to Decision Letter 1]

27 Nov 2020

Following the suggestion of Referee 2, we have modified a paragraph in the introduction, referring explicitly to Kermack-McKendrick seminal work. We have also explained that the “u” in uSEIR, refers to unitary, because our formulation makes it explicit that the inter-compartment transition rates conserve the total probability as time evolves. In order to improve the reading of the abstract, we have taken out the reference to uSEIR there.

---

## [Editor Report · Decision Letter 2]

3 Dec 2020

A new formulation of compartmental epidemic modelling for arbitrary distributions of incubation and removal times

PONE-D-20-15952R2

Dear Dr. Hernández,

We’re pleased to inform you that your manuscript has been judged scientifically suitable for publication and will be formally accepted for publication once it meets all outstanding technical requirements.

Kind regards,

Eric Forgoston

Academic Editor

PLOS ONE
---

## [Editor Report · Acceptance letter]

8 Jan 2021

PONE-D-20-15952R2 

A new formulation of compartmental epidemic modelling for arbitrary distributions of incubation and removal times 

Dear Dr. Hernández:

I'm pleased to inform you that your manuscript has been deemed suitable for publication in PLOS ONE. Congratulations! Your manuscript is now with our production department. 

Kind regards, 

on behalf of

Dr. Eric Forgoston 

Academic Editor

PLOS ONE